# Sodium Channel β Subunits—An Additional Element in Animal Tetrodotoxin Resistance?

**DOI:** 10.3390/ijms25031478

**Published:** 2024-01-25

**Authors:** Lorenzo Seneci, Alexander S. Mikheyev

**Affiliations:** 1Adaptive Biotoxicology Lab, School of the Environment, The University of Queensland, St Lucia, QLD 4067, Australia; uqlsenec@uq.edu.au; 2Evolutionary Genomics Group, Research School of Biology, Australian National University, Canberra, ACT 0200, Australia

**Keywords:** tetrodotoxin, resistance, sodium channels, subunit beta

## Abstract

Tetrodotoxin (TTX) is a neurotoxic molecule used by many animals for defense and/or predation, as well as an important biomedical tool. Its ubiquity as a defensive agent has led to repeated independent evolution of tetrodotoxin resistance in animals. TTX binds to voltage-gated sodium channels (VGSC) consisting of α and β subunits. Virtually all studies investigating the mechanisms behind TTX resistance have focused on the α subunit of voltage-gated sodium channels, where tetrodotoxin binds. However, the possibility of β subunits also contributing to tetrodotoxin resistance was never explored, though these subunits act in concert. In this study, we present preliminary evidence suggesting a potential role of β subunits in the evolution of TTX resistance. We gathered mRNA sequences for all β subunit types found in vertebrates across 12 species (three TTX-resistant and nine TTX-sensitive) and tested for signatures of positive selection with a maximum likelihood approach. Our results revealed several sites experiencing positive selection in TTX-resistant taxa, though none were exclusive to those species in subunit β1, which forms a complex with the main physiological target of TTX (VGSC Na_v_1.4). While experimental data validating these findings would be necessary, this work suggests that deeper investigation into β subunits as potential players in tetrodotoxin resistance may be worthwhile.

## 1. Introduction

Tetrodotoxin (hereafter TTX) is one of the most extensively studied and widely known natural toxins. More specifically, TTX is a small alkaloid with potent neurotoxic properties that is likely produced by bacteria, but it has been secondarily and independently sequestered by numerous species across the animal kingdom [1,2]. In fact, this toxin is found in fishes, newts, toads, and cephalopods, among others [1,2,3]. Most of these animals store TTX in specific glands for defensive use against predators, although some species—most notably, the blue-ringed octopuses (*Hapaloclaena* sp.)—employ it as a predatory venom to incapacitate prey [3,4].

This toxin is also a serious health hazard when ingested by humans, which is a serious concern in areas where TTX-laden pufferfish species feature in the local cuisine (e.g., Japan) [2]. Symptoms of TTX poisoning include nausea, vomiting, weakness, and hypotension, with systemic paralysis and subsequent respiratory failure causing fatalities in severe cases [2,5]. The neurotoxicity of TTX stems from its action as a sodium channel blocker, binding to the α-subunit of voltage-gated sodium channels (hereafter VGSCs) at neuromuscular and/or neural junctions to prevent the passage of Na^+^ ions into the channel pore by blocking it [5,6]. This prevents the transmission of neural input to skeletal muscles (including the diaphragm), which are therefore paralyzed.

Structurally, VGSCs are heteromeric protein complexes consisting of a central ~250 kDa α-subunit flanked by two or more ~40 kDa β-subunits [7,8,9]. The α subunit is further divided into four voltage-sensing domains (VSD I-IV) consisting of six hydrophobic transmembrane segments (S1–S6) [6,7,10]. They are normally located on the extracellular membrane, from which they allow the inflow of Na^+^ ions into the cell to trigger action potentials that result in a spike in cellular activity (e.g., muscle contraction) [6,7,8]. TTX itself was crucial in the quest towards understanding VGSCs’ structure and function, as identification of the terminal region of the channel pore was greatly facilitated by studying the binding of this toxin to the S5-S6 region (i.e., the pore-forming domain) of VSD I [5,6,11,12]. VGSCs are ubiquitous across the tree of life, with nine isoforms coded by as many genes documented in vertebrates and referred to by the nomenclature Na_v_. More specifically, Na_v_1.1–1.3 and Na_v_ 1.6 are most common in the central nervous system, whereas Na_v_1.7–1.9 are dominant in the peripheral nervous system [13,14]. On the other hand, Na_v_1.4–1.5 are the main isoforms in the skeletal and cardiac muscles, respectively [13,15,16].

Given their crucial involvement in neural transmission across multiple physiological systems, it is not surprising that VGSCs are the target of several toxins that critically impair homeostasis by disrupting their function. These toxins are broadly divided into two categories based on their mechanism of action: namely, pore blockers and gating modifiers [6]. Pore-blocking toxins, such as TTX itself, have their binding site on the VGSC pore-forming domain, thereby physically occluding it to stop the influx of Na^+^ ions into the cell by preventing depolarization of the channel [6,17]. Conversely, gating modifiers, such as α- and β-scorpion toxins, as well as most conotoxins from cone snails, bind to receptor sites located outside of the pore, thus altering the gating properties of the channel allosterically in order to suppress activation or inactivation of VGSCs [6,18].

Of the nine VGSC categories, only Na_v_1.5, Na_v_1.8, and Na_v_1.9 are inherently resistant to TTX binding [13], but multiple species have evolved TTX resistance in the skeletal muscle Na_v_1.4 channel as well [19,20,21]. This not only includes animals that store TTX in their bodies for defense, but also predators that have developed resistance to the toxin in order to subdue TTX-bearing prey. The most notable example of this is arguably the coevolution of TTX sequestration and TTX resistance in rough-skinned newts (*Taricha* sp.) and garter snakes (*Thamnophis* sp.) in western North America, which has surged as a textbook example of an evolutionary arms race between species [21,22,23,24].

Virtually all studies on the molecular mechanisms of TTX resistance have focused on the α subunit, where the binding site of the toxin is situated. However, VGSCs are actually complexes of an α and two β subunits, four types of which (β-1 to β-4, also commonly referred to as SCN1B to SCN4B) are found in vertebrates [8,25]. Of these, SCN1B is ubiquitous across all Na_v_ isoforms and is the only known β subunit for the skeletal muscle channel Na_v_1.4 (the main physiological target of TTX), while SCN2B and SCN3B are found in the central and peripheral nervous system as well as in cardiac myocytes [13]. Long thought to serve a merely auxiliary role in VGSC structure and function, β subunits have seen newfound research interest in the last two decades and are now known to play an important role in the modulation of VGSC activity and as cell adhesion molecules [25,26,27,28,29,30], with misfolding and malfunctions of β subunits being at the root of several neuropathic and cardiovascular diseases [16,26,27,28].

Interestingly, the activity of certain toxins from spiders (ProTx-II from the velvet tarantula *Thrixopelma pruriens*), scorpions (OD1 from the yellow Iranian scorpion *Odontobuthus doriae*), and cone snails (μ- and μO-conotoxins) against Na_v_1.2, Na_v_1.6, and/or Na_v_1.7 appears to be reduced in the presence of certain β subunits but enhanced by others [31,32,33,34,35]. While ProTx-II, OD1, and μO-conotoxins are gating modifiers and therefore more likely to be affected by β subunits (which are themselves known to regulate channel gating) [34,36], μ-conotoxins are pore blockers like TTX [37], thus supporting the influence of β subunits on pore-blocking properties. However, the binding site of TTX within the channel pore differs from (although it overlaps with) that of μ-conotoxins [6], which likely explains why TTX itself and the nearly identical saxitoxin (STX) were seemingly unaffected by the presence of β subunits [14,31]. Nonetheless, this was not tested in sodium channels from any TTX-resistant species, and affinity for toxins is known to vary even within the same VGSC isoform across different organisms, as evidenced by the considerably stronger effects of cone snail toxin on the mouse Na_v_1.2 compared to its homolog in rats [32].

In this study, we tested for signatures of positive diversifying selection and evolutionary convergence across the four β subunit types in a sample of both TTX-sensitive and TTX-resistant vertebrates. We interpreted amino acid changes unique to TTX-resistant species as potential indicators of resistance to the toxin. While undoubtedly limited and preliminary, this work is, to our knowledge, the first investigation into the possible role of VGSC β subunits in the evolution of TTX resistance in animals.

## 2. Results

Globally, all SCNB sequences were characterized by strong purifying selection, with only the foreground branches (i.e., the TTX-resistant species, see Table 1) showing a minor proportion of sites likely to experience diversifying selection (Table 2, Appendix A). One site likely under significant positive selection was observed in SCN1B (Figure 1, Table 2)—namely, a G33N mutation exclusive to *T. rubripes*. However, this site is not part of the interacting surface between β1 and the Na_v_1.4 α subunit, nor is it located in proximity to the channel pore, where TTX binds (Figure 2). In contrast, a highly significant signature common to all TTX-resistant species was observed in SCN2B at position 35 of the alignment (Figure 3, Table 2), where the proline found in most TTX-sensitive taxa is replaced by a valine in *T. elegans* and *T. sirtalis* and a serine in *T. rubripes* (asparagine is found in the same position in the TTX-sensitive *E. electricus*). Sites 93 and 197 were identified as likely under significant positive selection in SCN3B, from which, however, *T. sirtalis* was excluded due to its excessively fragmentary sequence (Figure 4). Lastly, codeml did not detect any signatures of positive selection in SCN4B. It is important to note that while signal peptides were trimmed for analysis in codeml, site nomenclature follows the previous literature, which starts from the signal peptide for better comparability and consistency. Because SCNB transcripts vary in length across species (Table 2), position numbers follow the *H. sapiens* isoform for each subunit.

To estimate evolutionary convergence across TTX-resistant taxa, we inferred the ratio of nonsynonymous to synonymous combinatorial substitutions (i.e., nucleotide replacements shared across two or more branches of a phylogenetic tree) expressed by the metric ω_C_ [38] (see Section 4). Values of ω_C_ were far below the ω_C_ ≥ 3 threshold for all subunits across the three categories of evolutionary convergence estimated using CSUBST (Table 3). The same was true for the total posterior probability of amino acid convergence (O^C^_N_) available in the Appendix A. This pattern was not limited to convergence, as divergent substitutions (e.g., a shared amino acid mutating to two different residues across separate proteins) showed comparably low values as well (Appendix A).

## 3. Discussion

Our findings provide some preliminary evidence of positive selection for TTX resistance in vertebrate VGSC β subunits, though there are very few changes. This is not surprising, because VGSCs play a vital role in the transmission of neural input [7,8], and any change is therefore likely to be detrimental to their efficiency, be it in the α or β subunits. TTX resistance of α subunits is accomplished by minute, sequence-level changes [23,24,39,40]. The quantification of convergent evolution signatures inferred with CSUBST provides little support for convergence in VGSC β subunits of TTX-resistant taxa. Combined with the equally low values for other combinatorial substitution metrics and posterior probabilities of convergence, this strongly suggests that the development of TTX resistance in *Thamnophis* sp. and *T. rubripes* was not influenced by convergently evolved changes in VGSC β subunits, though lineage-specific adaptations are possible. This was expected given the biochemical features of β subunits, which regulate channel gating and Na^+^ current intensity in VGSC via binding outside of the channel pore [25,26,27,41]. Thus, pore-blocking toxins like TTX are unlikely to interact directly with β subunits, which are instead known to affect binding of gating modifier toxins that impair VGSC function allosterically [31,32,33].

Branch- and site-specific signatures of selection estimated in PAML reported a prevalence of negative purifying selection across all SCNB sequences in the one-ratio model of evolution, which indicates strongly conserved protein structures (Appendix A). While the two-ratio models with TTX-resistant taxa as the foreground lineage did reveal certain sites with significant positive selection signatures, the link with resistance to the toxin is far from conclusive. For instance, while the G33N substitution in SCN1B is exclusive to the TTX-resistant *T. rubripes*, it does not extend to the equally resistant garter snakes, which instead share the G33 variant found in all TTX-sensitive species. However, this does not conclusively rule out a potential role of N33 in TTX resistance for *T. rubripes* per se. In fact, replacing glycine with asparagine leads to a change in polarity because the former is nonpolar while the latter is a noncharged polar amino acid, which is therefore able to form hydrogen bonds that might cause changes in interactions with other proteins.

Because β1 is known to bind noncovalently to all TTX-sensitive channels, including the skeletal muscle isoform Na_v_1.4, which is the main physiological target of TTX [2,13,15], and modify the channel surface charge when present [42], it is possible that polarity alterations in its binding domain may lead or respond to changes in the α+β complex conformation, potentially affecting TTX binding. The crystal structures of human, electric eel, and American cockroach (*Periplaneta americana*) Na_v_1.4 in complex with β1 have been recently elucidated, with detailed descriptions of the interacting regions between the two subunits [15,17,43]. While position 33 is part of the Ig loop that docks onto the extracellular loops L5_I_ and L6_IV_ as well as S1 and S2 in VSDIII, this residue is not among those directly implicated in binding to the α subunit [15,43]. Moreover, the TTX-binding site in VGSC is located within the channel pore [6] and is therefore unlikely to be directly impacted by β1 coupling. Combined with the apparent enhancement of pore blockage by μ-conotoxin (which competes with TTX for binding to the channel pore) in Na_v_1.7 in the presence of β1 [31], overall, the available evidence points against a role of this particular subunit in conferring toxin resistance. However, the readily available Na_v_1.4 α subunit sequences from a multitude of taxa (both TTX-sensitive and TTX-resistant) and the extensive research work conducted on this channel in relation to TTX binding offer a venue for in silico and/or in vitro studies of the interaction between TTX and the Na_v_1.4 + β1 complex. With recent publications highlighting surprising results, such as novel substitutions conferring TTX resistance [44] and a lack of correlation between TTX resistance and polar contacts between the toxin and Na_v_1.4 [45], this channel and subunit β1 would be the most reasonable candidates for experimental investigation of any β subunit involvement in TTX resistance.

A similar but larger-scale mutation was observed in SCN2B, where the P35V/P35S substitutions extended to all TTX-resistant taxa (*Thamnophis* sp. and *T. rubripes*, respectively) and were not observed in any TTX-sensitive species. The P6V replacement in T. elegans and T. sirtalis may alter the binding affinity of the channel to TTX due to conformational changes induced by the replacement of rotationally restricted proline [46], although no experimental evidence of this is available. In addition, the P35S substitution in *T. rubripes* also entails a polar/nonpolar contribution due to the serine residue. However, position 35 is not involved in the covalent bond between β2 and the α subunit, which instead relies on other residues further downstream (e.g., C55, Y56, and R135 for Na_v_1.2) [35,47]. Nonetheless, this position is part of the Ig region that docks onto the α-subunit extracellular region and might therefore affect channel conductance and modulation in unexplored ways. From a physiological perspective, β2 is known to associate with various TTX-sensitive Na_v_ isoforms, particularly those found in the central and peripheral neurons [13,35,48]. Thus, it is, in theory, possible that species storing and employing TTX as a defense measure (e.g., pufferfishes and newts) might rely on a set of molecular adaptations to avoid autotoxicity in their nervous system that could involve β2 subunits as well. A larger sample size with more TTX-sensitive and TTX-resistant species from across the tree of animal life would help ascertain whether P35X mutations are indeed widespread in and/or exclusive to TTX-resistant taxa regardless of phylogenetic affinity and thus possibly involved in the evolution of TTX resistance. On the other hand, the P35A mutation found in *E. electricus*, although possibly significant in structural terms due to the unique properties of proline vs. alanine [46], is highly unlikely to confer TTX resistance because current evidence points to VGSCs in this species being TTX-sensitive. Given that alanine and valine are scarcely different in structural terms, it is also ultimately unlikely that a P35V mutation alone would promote TTX resistance in *Thamnophis* either.

On the other hand, the substitutions highlighted as likely evolving under strong positive selection at position 93 in SCN3B appeared to be either snake-specific (Q93E, observed in *T. elegans* and *P. textilis*) or single-species mutations (Q93T for *D. rerio*) rather than selective changes associated with TTX resistance. The same can be hypothesized for position 174 in the same subunit type, with another seemingly snake-specific substitution (D197N). Although the TTX-resistant *T. rubripes* presents a unique mutation at both positions (Q94F and D198F, which correspond to positions 93 and 197 in *H. sapiens* in our alignment), the latter is part of the intracellular domain and therefore not exposed to TTX (although it does interact with the α subunit) [27], whereas the former is not known to play any significant part in interactions with other molecules or folding [30,49]. However, the different charge, size, and polarity of phenylalanine (nonpolar) compared to glutamine (charged, polar) at position 93 might affect the tertiary structure of the protein, with two N-glycosylation sites (N99 and N102) located in close proximity to it [27]. Nonetheless, it remains unlikely that this substitution contributes to conferring TTX resistance to VGSCs.

The absence of sites likely experiencing positive selection in SCN4B among TTX-resistant species is rather surprising, as subunit β4 is known to bind to multiple Na_v_ channels and interfere with toxin binding (although not in the case of TTX itself) [27,33]. This indicates that β4 is unlikely to be involved in TTX resistance, although our limited evidence is not sufficient to draw conclusions. For instance, TTX-resistant species could present different combinations of α and β VGSC subunits compared to sensitive taxa, which might reduce the affinity of TTX to its normal targets.

Given that virtually all research on TTX resistance has focused on α subunits [20,21,23,39] and the vast majority of TTX toxicity studies have made use of non-resistant models (e.g., mice, rats) [31,33,48,50], determining the structure of the α + β complex of TTX-resistant species offers a rich venue for future research. In fact, our results are only a preliminary indication of the evolution of SCNB subunits in vertebrates, with only three TTX-resistant species sampled due to lack of available sequences for many other taxa (e.g., *Hapaloclaena* sp., *Taricha* sp.). The rampant spread and consistently decreasing cost of next-generation sequencing techniques will hopefully allow for complete sequencing of whole genomes or at least VGSC genes of more TTX-resistant species in the near future. Furthermore, evolutionary convergence is neither a prerequisite nor a sufficient condition for the evolution of toxin resistance either [45], as evidenced by the TTX-resistant species included in this study. In fact, while certain substitutions in the Na_v_1.4 α subunit known to confer resistance to TTX are shared across garter snakes and pufferfishes, others are unique to each lineage [20,24,44]. Thus, the low convergence metrics we described in this study do not rule out a potential involvement of β subunits in conferring TTX resistance, pending experimental work.

Lastly, even if sodium channel β subunits are conclusively proven not to directly interfere with TTX binding, research into unexplored pathways towards resistance to this toxin in animals beyond the well-known α subunit structural alterations should not be neglected. Sequestration of toxins by free-ranging molecules that prevent them from reaching their targets in the first place is likely more widespread than commonly acknowledged, as evidenced by recent work on autoresistance to the potent alkaloid batrachotoxin (BTX) in poison dart frogs and pitohui birds [51].

Moreover, toxin resistance need not be molecular or chemical in nature. Behavioral resistance to poisons and venoms is documented in several predators of toxic animals, such as secretary birds (*Sagittarius serpentarius*) forcefully stomping venomous snakes on the head to prevent bites [52] and Australian rakalis (*Hydromys chrysogaster*) methodically excising the venter of cane toads (*Rhinella marina*) to avoid their poison glands [53]. Despite the evident limitations of our investigation, we hope it will further incentivize interest in alternative routes to the evolution of toxin resistance in animals.

## 4. Materials and Methods

### 4.1. Sequence Alignment and Curation

mRNA sequences for sodium channel β subunit (hereafter SCNB) were sourced from GenBank via the Orthologs function. The sample of TTX-resistant species comprised one pufferfish (*Takifugu rubripes*) and two garter snakes (*Thamnophis elegans* and *T. sirtalis*), whereas the pool of TTX-sensitive species consisted of one teleost fish (*Electrophorus electricus*), two mammals (*Homo sapiens* and *Mus musculus*), one bird (*Gallus gallus*), two anuran amphibians (*Xenopus tropicalis* and *Bufo bufo*), one lizard (*Anolis carolinensis*), and one snake (*Pseudonaja textilis*). Sequences were selected to provide comprehensive phylogenetic coverage across the vertebrate radiation. The sequences were aligned with MUSCLE implemented in Aliview v. 1.1 and manually curated to adjust them to the correct reading frame and delete signal peptides as delimited in Genbank and/or Uniprot. Where signal peptide information was unavailable, a sequence was trimmed to align with others for which the region was clearly defined in the database of origin. A phylogenetic tree grouping the aforementioned species was retrieved from Timetree (timetree.org).

### 4.2. Testing for Signatures of Positive Selection

To detect signatures of selection, the SCNB datasets were run in the codeml program from the Phylogenetic Analysis using Maximum Likelihood (PAML) software (version 4.10.5) [54]. This program uses a PHYLIP alignment file and a phylogenetic tree in NEWICK format to evaluate the ratio of non-synonymous to synonymous nucleotide substitutions in each codon (dN/dS, expressed by the variable ⍵) as a measure of selection acting on a given amino acid sequence. More specifically, ⍵ < 1 indicates negative or purifying selection, ⍵ = 1 represents neutral evolution, and ⍵ > 1 marks positive or diversifying selection. The results are expressed in terms of posterior probability of a site and/or branch undergoing positive selection with a 95% significance threshold. As a preliminary test of positive selection in TTX-resistant vs. TTX-sensitive species across each entire alignment, the one-ratio model M0 was chosen to obtain a universal ω value for each category. Subsequently, variation in ⍵ both at the branch and site level (i.e., allowing for different selection regimes across branches of the tree as well as across sites within sequences) was assessed via the M2 branch–site model of evolution. For both analyses, the three TTX-resistant species were selected as the foreground branch, whereas the remaining nine TTX-sensitive species represented the background branch.

### 4.3. Testing for Signatures of Convergent Evolution

Convergence in amino acid substitutions was assessed through the ω_c_ metric devised by Fukushima and Pollock [38]. This parameter builds on the well-established ω variable representing dN/dS ratios to quantify the ratio of non-synonymous to synonymous convergence (dN_C_/dS_C_) occurring convergently across branches of the phylogenetic tree. ω_c_ was estimated in the Python program CSUBST (GitHub–kfuku52/csubst: Molecular convergence detection) using the same sequence alignments and phylogenies (in FASTA and Newick format, respectively) as in Section 4.2. *T. sirtalis*, *T. elegans*, and *T. rubripes* were specified as foreground branches in a separate regex file. ω_C_ values were reported for the three combinatorial substitution categories indicating convergence. These consist of standard convergence (substitution from any amino acid to a specific amino acid across multiple branches), discordant convergence (different ancestral amino acids mutating to a common specific amino acid across branches), and congruent convergence (a shared ancestral specific amino acid mutating to another across branches) [38].

#### D Protein Reconstruction and Labelling

Structural models of subunits β1 and β2 were retrieved from AlphaFold (www.alphafold.com, accessed on 6 December 2023) [55,56] and saved as .pdb files, which were subsequently visualized in UCSF Chimera v. 1.16 [57]. Positions of interest were labelled onto the sequence in Chimera and then further edited in Adobe Photoshop for better clarity. 

## Figures and Tables

**Figure 1 ijms-25-01478-f001:**
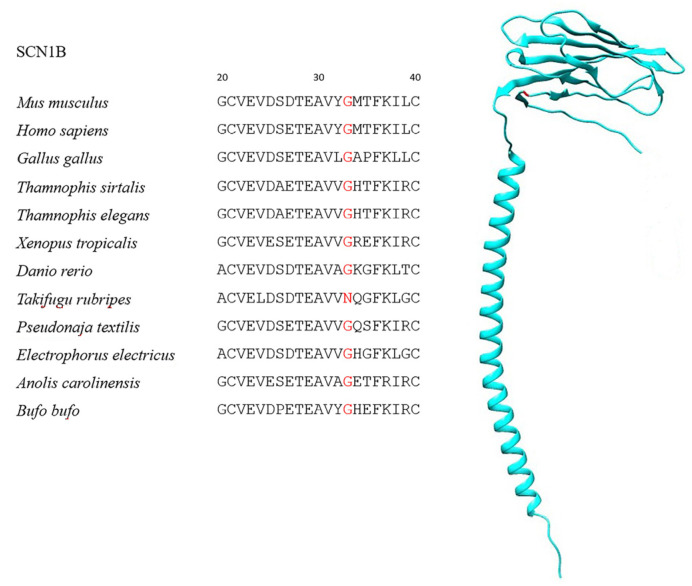
SCN1B sequences showing the positively selected G33N substitution in *T. rubripes* and the location of the residue on the protein (in red).

**Figure 2 ijms-25-01478-f002:**
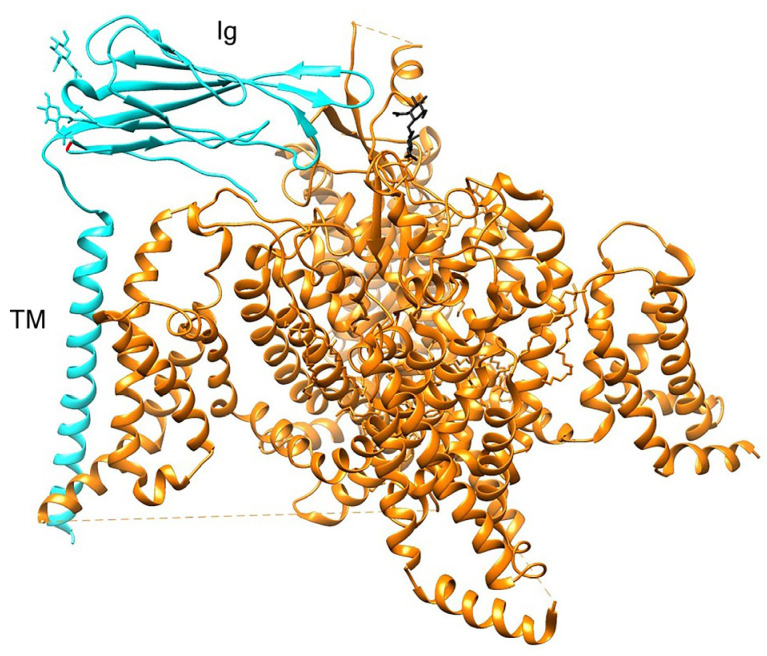
Crystal structure of the human Na_v_1.4 VGSC (orange) in complex with subunit β1 (cyan). TM = Transmembrane domain, Ig = Immunoglobulin domain. Residue G33 (mutated to N33 in the TTX-resistant *T. rubripes*) is highlighted in red. Structure retrieved from the Protein Data Bank following Pan et al. [15].

**Figure 3 ijms-25-01478-f003:**
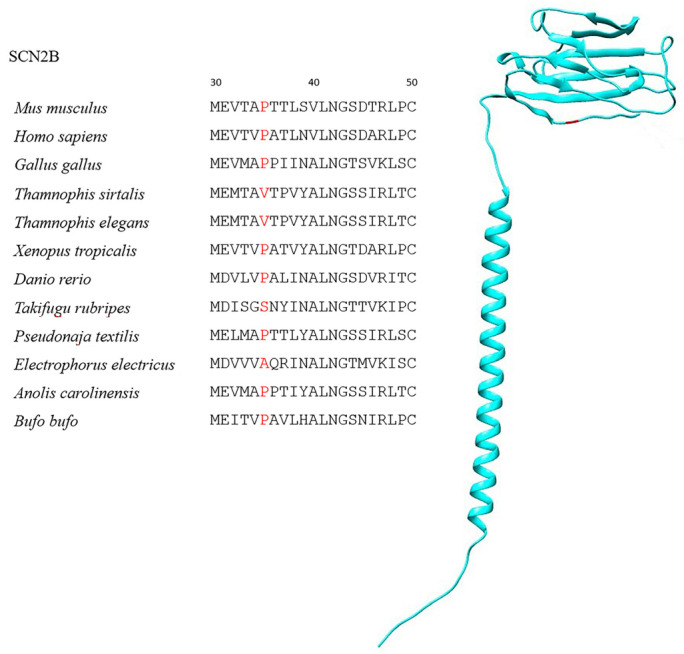
SCN2B sequences showing the positively selected P35X substitution in TTX-resistant taxa and *E. electricus* and the location of the residue on the protein (in red).

**Figure 4 ijms-25-01478-f004:**
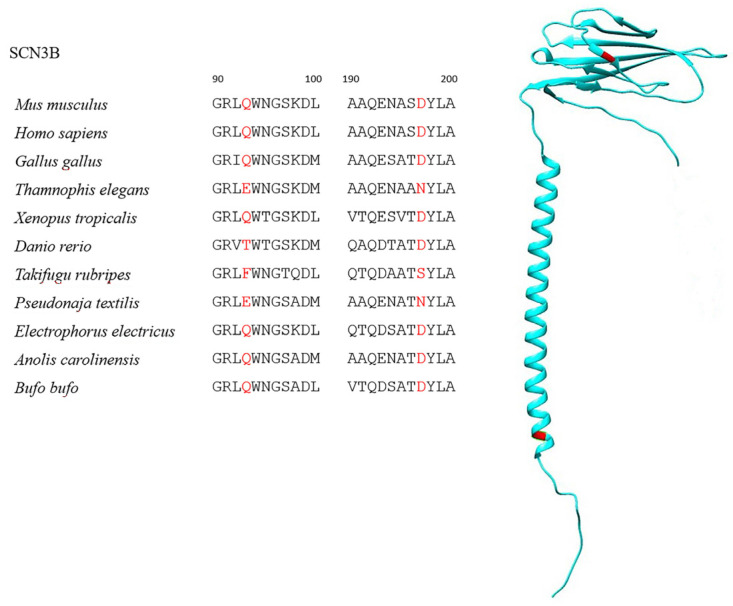
SCN3B sequences showing the positively selected Q93X and D197X substitutions in TTX-resistant taxa as well as *P. textilis* and *D. rerio*. The location of the residues on the protein is highlighted in red.

**Table 1 ijms-25-01478-t001:** List of species whose SCNB sequences were used in this study.

Species	Class	TTX-Resistant
*Homo sapiens*	Mammal	No
*Mus musculus*	Mammal	No
*Gallus gallus*	Bird	No
*Anolis carolinensis*	Reptile	No
*Thamnophis sirtalis*	Reptile	Yes
*Thamnophis elegans*	Reptile	Yes
*Pseudonaja textilis*	Reptile	No
*Bufo bufo*	Amphibian	No
*Xenopus tropicalis*	Amphibian	No
*Takifugu rubripes*	Fish	Yes
*Danio rerio*	Fish	No
*Electrophorus electricus*	Fish	No

**Table 2 ijms-25-01478-t002:** List of sites most likely under significant positive selection for all SCNB subunits as determined through codeml. Site numbers refer to mature protein sequences (i.e., without the signal peptide) from *H. sapiens*.

	Maximum Sequence Length (without Signal Peptide)	Positions under Positive Selection	Substitutions	Prob (ω > 1)
SCN1B	201	15	G33N	95.6%
SCN2B	197	6	P35V, P35S	99.5%
SCN3B *	199	70	Q93E, Q93F, Q93T	98.6%
		174	D197N, D197S	99.2%
SCN4B	208	-	-	-

* *T. sirtalis* not included due to excessively fragmented sequence.

**Table 3 ijms-25-01478-t003:** ω_C_ values for three modes of convergent substitution as obtained from CSUBST for TTX-resistant clades. See Section 4 for descriptions of all substitution categories. ω_C_ ≥ 3 threshold considered significant [38].

Combinatorial Substitution Categories	SCN1B	SCN2B	SCN3B *	SCN4B
**Convergence**	0.019	0.004	0.108	0.088
**Discordant Convergence**	0.023	0.001	0.095	0.752
**Congruent Convergence**	0.020	0.008	0.134	0.088

* *T. sirtalis* not included due to excessively fragmented sequence.

## Data Availability

All data are available in the Appendix A.

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
