# Peer review of "Sodium Channel β Subunits—An Additional Element in Animal Tetrodotoxin Resistance?"

_ijms, 2024, doi:10.3390/ijms25031478_

Round 1
Reviewer 1 Report
Comments and Suggestions for Authors
Seneci and Mikheyev have carried out a comparative sequence analysis of the beta subunit genes of the voltage-gated sodium channel family in species exhibiting differing degrees of TTX sensitivity. The rationale being that evolutionary pressure may have contributed to the emergence of variants which confer TTX resistance within the heteromeric complex, particularly with respect to Nav1.4. The topic is interesting and lends merit to any future experimental evaluation of potential β-subunit influence on TTX sensitivity, be that direct biophysical effects or roles in toxin sequestration. However there are several issues which need to be addressed to provide sufficient completeness and appropriately draw any preliminary conclusion. as to the potential role of beta evolution in enhanced TTX resistance.
· Sequence alignments ignore the signal peptide and it is not clearly established that the sequence alignments start at the mature protein. This is particularly confusing when comparing with the known reference sequence and other literature cited. Despite this region being absent from the mature protein, it is apparent in the alpha fold structures in Fig 1 and Fig 2. Furthermore, the corresponding alignments for SCN3B and its structure are not shown. It is not clear why this has been left out and only adds to the confusion, requiring the reader to make inferences about the location of these residues. Presumably, D185N and D185S are in fact D197. Q81F is less clear, it is either Q93 or Q101, but neither corresponds to the 13 amino acid difference identified between D185 and D197 suggesting some variation in the intervening sequence.
· The pdb structure of Nav1.4 and beta 1 is available. It would be illuminating to see the positioning of the beta subunit mutations highlighted on this structure. The authors point out that the Ig domain is involved in mediating the alpha beta interaction, however it is not clear that the G15N (G33N in the complete sequence?) is within this binding interface. The ability to form H-bonds may cause disruption in terms of higher order complexes or further interacting proteins……
· The mutation P6D (aspartic acid) is mentioned in line 182 as occurring in E. electricus. Yet the sequence alignment in fig 2 shows an A (alanine). Furthermore in the rest of the sentence when describing the significance of the mutation it has become an asparagine. In the full beta 2 sequence this residue presumably is P35? This might be quite an interesting residue for functional effects based on the orientation of the beta 2 subunit in the Nav1.7 - structure…… In line 169 – 171 the position of this residue as P6 is considered in relation to C55, Y56 and R135, but the nomenclature is not consistent i.e. these residues may be in closer proximity than suggested. Again a figure to highlight some of the discussion would be beneficial.
· Potential glycosylation of the G15N residue is suggested, however, this residue does not fit the consensus motif for glycosylation (N-X-Ser/Thr).
· The authors state that the intracellular domain of beta 3 (line 192-193) would not be exposed to alpha subunit or toxin. While true for the latter, it is not the case for the alpha subunit which has been shown to interact with the beta subunit via the intracellular domains. Indeed, the cited ref (25 – Namadurai et al.) makes this very point. Also see Spampanto J et al 2004.
· In SCN3B the variant identified at position 81 is discussed in the context of its proximity to N85 and N89 highlighted in ref 25. But, as before, these numbering formats are not consistent.
· Reference is made to previous reports of evolved TTX resistance in Nav1.4 of certain species. Is there any known correlation between the evolution of Nav1.4 in the TTX resistant species assessed here? Specifically within the TTX binding site.
· Given that the beta subunit binding sites are distinct from the TTX binding site on the Nav1.5, it is hard to see how the beta subunits would be expected to directly impact the binding of TTX. Some discussion of this should be included.
· The beta subunit genes have been referred to as SCNB1, SCNB2 etc… they are in fact SCN1B, SCN2B etc.
· Line 20 – 21 this sentence isn’t clear
· Line 123 – 124 the supplementary material does not provide transparent representation of this. A table/figure to highlight the point would be helpful to the reader.
· Line 159 – 159. What evidence is there for beta 1 to ‘most notably’ bind the skeletal muscle Nav1.4? ‘Including’ may be a more appropriate term here.
Author Response
We are grateful to the Reviewer for a wide range of constructive suggestions. We felt that we were able to address them all in the revision and they significantly improved the manuscript. The responses are reported below.
Sequence alignments ignore the signal peptide and it is not clearly established that the sequence alignments start at the mature protein. This is particularly confusing when comparing with the known reference sequence and other literature cited. Despite this region being absent from the mature protein, it is apparent in the alpha fold structures in Fig 1 and Fig 2. Furthermore, the corresponding alignments for SCN3B and its structure are not shown. It is not clear why this has been left out and only adds to the confusion, requiring the reader to make inferences about the location of these residues. Presumably, D185N and D185S are in fact D197. Q81F is less clear, it is either Q93 or Q101, but neither corresponds to the 13 amino acid difference identified between D185 and D197 suggesting some variation in the intervening sequence.
Sequence position numbering was adjusted to match previous literature (i.e. starting from the signal peptide) and a clarifying statement was added in the Results section to assist the reader (lines 124-128).
The pdb structure of Nav1.4 and beta 1 is available. It would be illuminating to see the positioning of the beta subunit mutations highlighted on this structure. The authors point out that the Ig domain is involved in mediating the alpha beta interaction, however it is not clear that the G15N (G33N in the complete sequence?) is within this binding interface. The ability to form H-bonds may cause disruption in terms of higher order complexes or further interacting proteins……
The .pdb structure of human Nav1.4+beta1 from Pan et al. (2018) was retrieved from Protein Data Bank and processed for visualization as Figure 2, with specific discussion of the potential role of G33N in complex formation and TTX binding (lines 197-206).
The mutation P6D (aspartic acid) is mentioned in line 182 as occurring in E. electricus. Yet the sequence alignment in fig 2 shows an A (alanine). Furthermore in the rest of the sentence when describing the significance of the mutation it has become an asparagine. In the full beta 2 sequence this residue presumably is P35? This might be quite an interesting residue for functional effects based on the orientation of the beta 2 subunit in the Nav1.7 - structure…… In line 169 – 171 the position of this residue as P6 is considered in relation to C55, Y56 and R135, but the nomenclature is not consistent i.e. these residues may be in closer proximity than suggested. Again a figure to highlight some of the discussion would be beneficial.
Wording was fixed to identify the mutated amino acid in question as A (alanine) at position 35 in SCN2B (line 232).
Potential glycosylation of the G15N residue is suggested, however, this residue does not fit the consensus motif for glycosylation (N-X-Ser/Thr).
Mention of glycosylation for G33N was removed according to the Reviewer's recommendation.
The authors state that the intracellular domain of beta 3 (line 192-193) would not be exposed to alpha subunit or toxin. While true for the latter, it is not the case for the alpha subunit which has been shown to interact with the beta subunit via the intracellular domains. Indeed, the cited ref (25 – Namadurai et al.) makes this very point. Also see Spampanto J et al 2004.
The paragraph was modified to clarify that the intracellular domain of SCN3B does interact with the alpha subunit (lines 245-246).
In SCN3B the variant identified at position 81 is discussed in the context of its proximity to N85 and N89 highlighted in ref 25. But, as before, these numbering formats are not consistent.
For the numbering of SCN3B residues, see point 3 above.
Reference is made to previous reports of evolved TTX resistance in Nav1.4 of certain species. Is there any known correlation between the evolution of Nav1.4 in the TTX resistant species assessed here? Specifically within the TTX binding site.
A brief mention of convergent substitutions in the Nav1.4 alpha subunit was added in reference to the role of convergence in the development of TTX resistance (lines 270-274).
Given that the beta subunit binding sites are distinct from the TTX binding site on the Nav1.5, it is hard to see how the beta subunits would be expected to directly impact the binding of TTX. Some discussion of this should be included.
A discussion of the binding mechanism of beta subunits and their lack of direct interaction with TTX was added (lines 173-178).
The beta subunit genes have been referred to as SCNB1, SCNB2 etc… they are in fact SCN1B, SCN2B etc.
The SCNB subunit nomenclature was fixed to reflect the Reviewer's feedback.
Line 20 – 21 this sentence isn’t clear
The sentence was modified to improve clarity.
Line 123 – 124 the supplementary material does not provide transparent representation of this. A table/figure to highlight the point would be helpful to the reader.
The sentence was changed to assist the reader in interpreting the information (lines 154-157). The Supplementary Material includes an extended table with all values obtained from CSUBST, which we believe would be cumbersome and redundant to include in the main text. Furthermore, the relevant data for the sake of our analysis (i.e. those metrics that refer to convergent evolution specifically) are listed in Table 3, whereas other measures refer to distinct mechanisms of combinatorial substitutions such as divergent evolution.
Line 159 – 159. What evidence is there for beta 1 to ‘most notably’ bind the skeletal muscle Nav1.4? ‘Including’ may be a more appropriate term here.
This was adjusted according to the Reviewer's feedback (lines 191-192).
Reviewer 2 Report
Comments and Suggestions for Authors
The authors present an interesting question regarding the evolution of TTX resistance in sodium channels across taxa, namely that the auxiliary beta subunits might provide some selection pressure. Their results point to a few sites within several beta subunits where mutations could influence selection. The low sample size of available mRNA sequences does hinder any strong conclusions.
I have two possible suggestions for this work.
First, one of the mutations highlighted is G15N in SCN1B which associates with SCN4A skeletal muscle channel which has been extensively studied from many perspectives including structure to function, channelopathies, toxin effects, and structural determination. In fact the complex of the alpha and beta 1 subunits for the human skeletal sodium channel has been solved by cryo-EM and with bound TTX (Shen et al 2018; pdb 6A95). this study should be part of the reference list and discussed. Also, this structure offers the opportunity to test whether beta subunit 1 mutation affects TTX binding with a computational approach as in molecular dynamics or static docking studies. there are other references to structures determined for toxin binding that could be explored also. Homology modeling of snake channels with investigation of TTX binding has been done, with docking studies (Geffeney et al 2023).
Second, while other effects of beta subunits besides modulation of pore blockage which is not apparent from the results presented here are given in the discussion, I would suggest that a more careful examination of the studies presented in the Introduction lines 79-89 regarding toxin efficacy and beta subunits is in order. Specifically, toxins acting on sodium channels can be classified by several means, and one of these is mechanism of action. Here, the most general distinction is toxins that act as pore blockers versus toxins that act as gating modifiers (by "general" I mean this is a very simplified way to look at toxins acting on VGSCs but useful here). For example, protoxin is mentioned as one whose action can be modified by beta subunit, and "cone snail toxin" is another. Protoxin II is a gating modifier toxin (not a pore blocker). Some cone snail toxins such as the uO and delta toxins also produce their effect by modifying gating. Whereas u-conotoxins, TTX and STX act as pore blockers. Studies on the functional impact of beta subunits have largely concluded that they modify the kinetics of channel state transitions. Mutations in beta subunits could have many effects including channel clustering, trafficking etc - but if one were to hypothesize whether mutations in beta subunits affect sodium channels by acting on the pore module, or acting on the voltage sensor module, it might be a good first step to make that distinction in your writing of this section regarding these toxins. They are not equivalent functionally, and cone snail toxins acting on VGSCs comprise several families that are also non equivalent functionally. There are only few studies here on beta subunits and toxin binding as you have seen, but the observation that beta subunit functional characterization shows them to modify gating is consistent with the studies showing that TTX binding is unaffected by presence of beta subunit. Thus I would suggest that this section be more specific in its presentation please, and additional references overall might be warranted.
Author Response
We are grateful to the Reviewer for a wide range of constructive suggestions. We felt that we were able to address them all in the revision and they significantly improved the manuscript. Our responses are reported below.
First, one of the mutations highlighted is G15N in SCN1B which associates with SCN4A skeletal muscle channel which has been extensively studied from many perspectives including structure to function, channelopathies, toxin effects, and structural determination. In fact the complex of the alpha and beta 1 subunits for the human skeletal sodium channel has been solved by cryo-EM and with bound TTX (Shen et al 2018; pdb 6A95). this study should be part of the reference list and discussed. Also, this structure offers the opportunity to test whether beta subunit 1 mutation affects TTX binding with a computational approach as in molecular dynamics or static docking studies. there are other references to structures determined for toxin binding that could be explored also. Homology modeling of snake channels with investigation of TTX binding has been done, with docking studies (Geffeney et al 2023).
The study by Pan et al. (2018) was extensively referenced in the main text and the .pdb file of their Nav1.4+beta1 figure was retrieved to be processed as Figure 2. While we agree with the Reviewer that docking models showing TTX binding to the complex itself would be highly informative, none were constructed due to limited time and resources, as well as it being beyond the scope of this preliminary investigation. The reference provided by the Reviewer (Geffeney et al. 2023) was nonetheless added and discussed in the manuscript.
Second, while other effects of beta subunits besides modulation of pore blockage which is not apparent from the results presented here are given in the discussion, I would suggest that a more careful examination of the studies presented in the Introduction lines 79-89 regarding toxin efficacy and beta subunits is in order. Specifically, toxins acting on sodium channels can be classified by several means, and one of these is mechanism of action. Here, the most general distinction is toxins that act as pore blockers versus toxins that act as gating modifiers (by "general" I mean this is a very simplified way to look at toxins acting on VGSCs but useful here). For example, protoxin is mentioned as one whose action can be modified by beta subunit, and "cone snail toxin" is another. Protoxin II is a gating modifier toxin (not a pore blocker). Some cone snail toxins such as the uO and delta toxins also produce their effect by modifying gating. Whereas u-conotoxins, TTX and STX act as pore blockers. Studies on the functional impact of beta subunits have largely concluded that they modify the kinetics of channel state transitions. Mutations in beta subunits could have many effects including channel clustering, trafficking etc - but if one were to hypothesize whether mutations in beta subunits affect sodium channels by acting on the pore module, or acting on the voltage sensor module, it might be a good first step to make that distinction in your writing of this section regarding these toxins. They are not equivalent functionally, and cone snail toxins acting on VGSCs comprise several families that are also non equivalent functionally. There are only few studies here on beta subunits and toxin binding as you have seen, but the observation that beta subunit functional characterization shows them to modify gating is consistent with the studies showing that TTX binding is unaffected by presence of beta subunit. Thus I would suggest that this section be more specific in its presentation please, and additional references overall might be warranted.
The distinction between pore-blocking and gating modifier toxins was described in the introduction to implement the Reviewer's feedback (lines 61-68). Better clarification of the activities of ProTx-II, OD1, and m-conotoxins vs TTX was included as well (lines 93-96). This was then further appraised in the discussion in the context of the lack of convergence we report for SCNB sequences in this study (lines 173-178).
Reviewer 3 Report
Comments and Suggestions for Authors
The authors investigated a possible correlation between Nav beta subunit mutations and TTX resistance in several TTX sensitive and resistant species. Although they found some potentially relevant hits, their findings are no more than a working hypothesis. At the most interesting location position 6 SCNB2, is not exclusive to TTX resistant species since a P to A mutation (this is alanine according to Fig. 2, in contrast to asparagine as mentioned in the text) is found in the TTX sensitive Electrophorus electricus. Since an alanine residue is structurally not very different to the valine residue found in the TTX resistant species Thamnophis sirtalis and Thamnophis elegans, a relevance for TTX sensitivity seems to be not very high.
To demonstrate a relevance of their findings, the authors must model the effect of the found mutations on the SCNB2 and even better, on the Nav alpha subunit which binds SCNB2.
Also, the authors must investigate, if there are any mutations on the Nav alpha subunits putatively responsible for a deviation in the TTX sensitivity.
Furthermore, at least one experiment should be shown where a supposed beta subunit mutation leading to TTX resistance is effective in preventing block of Nav channel alpha subunits of TTX sensitive species.
Author Response
We are grateful to the Reviewer for a wide range of constructive suggestions. We felt that we were able to address them all in the revision and they significantly improved the manuscript. Our responses are reported below.
The authors investigated a possible correlation between Nav beta subunit mutations and TTX resistance in several TTX sensitive and resistant species. Although they found some potentially relevant hits, their findings are no more than a working hypothesis. At the most interesting location position 6 SCNB2, is not exclusive to TTX resistant species since a P to A mutation (this is alanine according to Fig. 2, in contrast to asparagine as mentioned in the text) is found in the TTX sensitive Electrophorus electricus. Since an alanine residue is structurally not very different to the valine residue found in the TTX resistant species Thamnophis sirtalis and Thamnophis elegans, a relevance for TTX sensitivity seems to be not very high.
We agree with the Reviewer's point that the two amino acids are similar and have adjusted the text accordingly (lines 235-237). The wording was also changed to identify the mutated amino acid as alanine rather than asparagine.
To demonstrate a relevance of their findings, the authors must model the effect of the found mutations on the SCNB2 and even better, on the Nav alpha subunit which binds SCNB2.
Also, the authors must investigate, if there are any mutations on the Nav alpha subunits putatively responsible for a deviation in the TTX sensitivity.
Furthermore, at least one experiment should be shown where a supposed beta subunit mutation leading to TTX resistance is effective in preventing block of Nav channel alpha subunits of TTX sensitive species.
While we agree with the Reviewer that in silico and in vitro experiments are required to determine whether beta subunits play an active role in TTX resistance, this is beyond the scope of our preliminary investigation (as we have clarified in the text). Furthermore, such experiments would require a substantial investment of time and resources that we are currently unable to undertake.
Round 2
Reviewer 1 Report
Comments and Suggestions for Authors
We are grateful to the Reviewer for a wide range of constructive suggestions. We felt that we were able to address them all in the revision and they significantly improved the manuscript. The responses are reported below.
Sequence alignments ignore the signal peptide and it is not clearly established that the sequence alignments start at the mature protein. This is particularly confusing when comparing with the known reference sequence and other literature cited. Despite this region being absent from the mature protein, it is apparent in the alpha fold structures in Fig 1 and Fig 2. Furthermore, the corresponding alignments for SCN3B and its structure are not shown. It is not clear why this has been left out and only adds to the confusion, requiring the reader to make inferences about the location of these residues. Presumably, D185N and D185S are in fact D197. Q81F is less clear, it is either Q93 or Q101, but neither corresponds to the 13 amino acid difference identified between D185 and D197 suggesting some variation in the intervening sequence.
Sequence position numbering was adjusted to match previous literature (i.e. starting from the signal peptide) and a clarifying statement was added in the Results section to assist the reader (lines 124-128).
Table 2 is internally inconsistent. ‘Positions under positive selection’ and ‘substitutions’ do not match.
The sequence positioning is not noted in Fig 1. Fig3. And Fig 4. For example, in Fig 3 the SCN2B sequences displayed presumably start at position 30 given the location of the P35 residue, but this is being assumed. Please highlight the number of every 10th residue in all sequence alignments for clarity.
Fig 4. Is also not consistent with Fig 1 and Fig 3. It is shown as two short separate sequences rather than one continuous read. Why?
The alpha fold structures in figures 1, 3 and 4 are inaccurate as they still include the signal peptide which does not form part of the mature protein (as can be seen in fig 2).
The pdb structure of Nav1.4 and beta 1 is available. It would be illuminating to see the positioning of the beta subunit mutations highlighted on this structure. The authors point out that the Ig domain is involved in mediating the alpha beta interaction, however it is not clear that the G15N (G33N in the complete sequence?) is within this binding interface. The ability to form H-bonds may cause disruption in terms of higher order complexes or further interacting proteins……
The .pdb structure of human Nav1.4+beta1 from Pan et al. (2018) was retrieved from Protein Data Bank and processed for visualization as Figure 2, with specific discussion of the potential role of G33N in complex formation and TTX binding (lines 197-206).
Superimposed as text, the G33N residue label is hard to see. Please highlight the residue itself on the protein (i.e consistent with the approach to the alpha fold images in fig 1, 2 and 3). Why are some side chains highlighted in the Ig domain of the beta subunit and in the Nav1.4 structure (in black). Is this meant to indicate something that has not been explained?
The mutation P6D (aspartic acid) is mentioned in line 182 as occurring in E. electricus. Yet the sequence alignment in fig 2 shows an A (alanine). Furthermore in the rest of the sentence when describing the significance of the mutation it has become an asparagine. In the full beta 2 sequence this residue presumably is P35? This might be quite an interesting residue for functional effects based on the orientation of the beta 2 subunit in the Nav1.7 - structure…… In line 169 – 171 the position of this residue as P6 is considered in relation to C55, Y56 and R135, but the nomenclature is not consistent i.e. these residues may be in closer proximity than suggested. Again a figure to highlight some of the discussion would be beneficial.
Wording was fixed to identify the mutated amino acid in question as A (alanine) at position 35 in SCN2B (line 232).
Potential glycosylation of the G15N residue is suggested, however, this residue does not fit the consensus motif for glycosylation (N-X-Ser/Thr).
Mention of glycosylation for G33N was removed according to the Reviewer's recommendation.
The authors state that the intracellular domain of beta 3 (line 192-193) would not be exposed to alpha subunit or toxin. While true for the latter, it is not the case for the alpha subunit which has been shown to interact with the beta subunit via the intracellular domains. Indeed, the cited ref (25 – Namadurai et al.) makes this very point. Also see Spampanto J et al 2004.
The paragraph was modified to clarify that the intracellular domain of SCN3B does interact with the alpha subunit (lines 245-246).
In SCN3B the variant identified at position 81 is discussed in the context of its proximity to N85 and N89 highlighted in ref 25. But, as before, these numbering formats are not consistent.
For the numbering of SCN3B residues, see point 3 above.
Reference is made to previous reports of evolved TTX resistance in Nav1.4 of certain species. Is there any known correlation between the evolution of Nav1.4 in the TTX resistant species assessed here? Specifically within the TTX binding site.
A brief mention of convergent substitutions in the Nav1.4 alpha subunit was added in reference to the role of convergence in the development of TTX resistance (lines 270-274).
Given that the beta subunit binding sites are distinct from the TTX binding site on the Nav1.5, it is hard to see how the beta subunits would be expected to directly impact the binding of TTX. Some discussion of this should be included.
A discussion of the binding mechanism of beta subunits and their lack of direct interaction
The beta subunit genes have been referred to as SCNB1, SCNB2 etc… they are in fact SCN1B, SCN2B etc.
The SCNB subunit nomenclature was fixed to reflect the Reviewer's feedback.
Line 20 – 21 this sentence isn’t clear
The sentence was modified to improve clarity.
Line 123 – 124 the supplementary material does not provide transparent representation of this. A table/figure to highlight the point would be helpful to the reader.
The sentence was changed to assist the reader in interpreting the information (lines 154-157). The Supplementary Material includes an extended table with all values obtained from CSUBST, which we believe would be cumbersome and redundant to include in the main text. Furthermore, the relevant data for the sake of our analysis (i.e. those metrics that refer to convergent evolution specifically) are listed in Table 3, whereas other measures refer to distinct mechanisms of combinatorial substitutions such as divergent evolution.
Line 159 – 159. What evidence is there for beta 1 to ‘most notably’ bind the skeletal muscle Nav1.4? ‘Including’ may be a more appropriate term here.
This was adjusted according to the Reviewer's feedback (lines 191-192).
There are some fundamental misunderstandings of VGSCs. Lines 46 – 47 describe the channel as being divided into four voltage sensing domains (VSDs) consisting of 6 transmembrane regions. This is inaccurate. The VSDs consist of the first 4 transmembrane regions of each domain (DI-DIV). In line 52 – 53, the identification of the pore domain with the assistance of TTX is discussed as if it resides in the voltage sensing domain. The VSD and PD are separate transmembrane structures.
Lines 63 – 66. TTX binding occludes the pore stopping influx of Na+ which prevents membrane depolarization. A lack of depolarization is the consequence of pore occlusion not the mechanism for occlusion, and sodium channels do not themselves become depolarized.
Author Response
We are grateful to Reviewer 1 for their constructive feedback and useful advice, which we have implemented as follows.
Table 2 is internally inconsistent. ‘Positions under positive selection’ and ‘substitutions’ do not match.
This has been adjusted to match the nomenclature in the rest of the manuscript. The column showing the length of each subunit (in amino acid residues) was deleted to avoid confusion between residue nomenclature (which starts from the signal peptide) and total length values reported therein (which did not include the SP).
The sequence positioning is not noted in Fig 1. Fig3. And Fig 4. For example, in Fig 3 the SCN2B sequences displayed presumably start at position 30 given the location of the P35 residue, but this is being assumed. Please highlight the number of every 10th residue in all sequence alignments for clarity.
The figures were modified in line with the Reviewer's feedback (see next point for further clarification).
Fig 4. Is also not consistent with Fig 1 and Fig 3. It is shown as two short separate sequences rather than one continuous read. Why?
SCN3B showed two sites under significant positive selection, i.e. positions 93 and 197. Given the approximately 100-residue distance between the two, we deemed it impractical to show the entire interval that separates them in Fig. 3. Nonetheless, all figures were modified to include residues starting from the nearest 10th position upstream to the positively selected site and ending to the nearest 10th position downstream to it.
The alpha fold structures in figures 1, 3 and 4 are inaccurate as they still include the signal peptide which does not form part of the mature protein (as can be seen in fig 2).
The signal peptide was deleted in Adobe Photoshop.
Superimposed as text, the G33N residue label is hard to see. Please highlight the residue itself on the protein (i.e consistent with the approach to the alpha fold images in fig 1, 2 and 3). Why are some side chains highlighted in the Ig domain of the beta subunit and in the Nav1.4 structure (in black). Is this meant to indicate something that has not been explained?
The figure was modified in line with the Reviewer's feedback. The side chains represented as sticks indicate glycosyl moieties at known glycosylation sites, which was specified in the figure caption following Pan et al. (2018)
There are some fundamental misunderstandings of VGSCs. Lines 46 – 47 describe the channel as being divided into four voltage sensing domains (VSDs) consisting of 6 transmembrane regions. This is inaccurate. The VSDs consist of the first 4 transmembrane regions of each domain (DI-DIV). In line 52 – 53, the identification of the pore domain with the assistance of TTX is discussed as if it resides in the voltage sensing domain. The VSD and PD are separate transmembrane structures.
The sentence was modified and expanded in line with the Reviewer's feedback (lines 46-54, 58-61).
Lines 63 – 66. TTX binding occludes the pore stopping influx of Na+ which prevents membrane depolarization. A lack of depolarization is the consequence of pore occlusion not the mechanism for occlusion, and sodium channels do not themselves become depolarized.
The wording was modified in line with the Reviewer's feedback (lines 71-74)
Reviewer 3 Report
Comments and Suggestions for Authors
Unfortunately, the authors are not able to strengthen their hypothesis of a significant contribution of Nav beta subunits on the TTX sensitivity by additional experiments.
Therefore, their manuscript describes a pure hypothesis standing on shaky ground.
Author Response
We appreciate Reviewer 3's concern about the lack of experimental support. We seriously considered this approach and contacted several potential collaborators, who expressed reservations about the difficulty of conducting such work. Here is a representative quote: "a bottleneck is to obtain the Nav αβ coding genes from the actual TTX-R organisms to test, and having them working in our usual cell lines (e.g. HEK293, CHO). If these TTX-resistant organisms have similar α and β subunits with human or murine, we might be able to test these mutants on human or murine instead." While we have overcome the first bottleneck in this study by identifying the genes from TTX-R organisms, whether they would work in established cell lines remains uncertain. Similarly, they are divergent enough from human and murine homologs that there are no specific mutations representing obvious targets. Therefore, we have pushed this approach about as far as it would practically go, which is a computational analysis. Expecting more at this stage seems unrealistic, but we have qualified our conclusions accordingly.
Round 3
Reviewer 3 Report
Comments and Suggestions for Authors
I am satisfied with the decision to publish the manuscript as a hypothesis.
Comments on the Quality of English Languageno further comment